# Synthesis and Biological Evaluation of PEGylated MWO_4_ Nanoparticles as Sonodynamic AID Inhibitors in Treating Diffuse Large B-Cell Lymphoma

**DOI:** 10.3390/molecules27217143

**Published:** 2022-10-22

**Authors:** Junna Jiao, Zhuang Qian, Yurong Wang, Mei Liu, Liye Fan, Mengqing Liu, Zichen Hao, Junrong Jiao, Zhuangwei Lv

**Affiliations:** 1Department of Immunology, School of Basic Medical Sciences, Xinxiang Medical University, Xinxiang 453003, China; 2Xinxiang Key Laboratory of Tumor Vaccine and Immunotherapy, Xinxiang Medical University, Xinxiang 453003, China; 3Institutes of Health Central Plains, Xinxiang Medical University, Xinxiang 453003, China; 4Key Lab for Special Functional Materials of Ministry of Education, National & Local Joint Engineering Research Center for High-Efficiency Display and Lighting Technology, School of Materials Science and Engineering, Collaborative Innovation Center of Nano Functional Materials and Applications, Henan University, Kaifeng 475004, China; 5School of Forensic Medicine, Xinxiang Medical University, Xinxiang 453003, China

**Keywords:** activation-induced cytidine deaminase, inhibitors, PEGylated MWO_4_ nanoparticles, DLBCL treatment

## Abstract

Sonodynamic therapy (SDT) triggered by ultrasound (US) has attracted increasing attention owing to its ability to overcome critical limitations, including low tissue-penetration depth and phototoxicity in photodynamic therapy (PDT). Biogenic metal oxide nanoparticles (NPs) have been used as anti-cancer drugs due to their biocompatibility properties with most biological systems. Here, sonosensitizer MWO_4_-PEG NPs (M = Fe Mn Co Ni) were synthesized as inhibitors to activation-induced cytidine deaminase (AID), thus neutralizing the extensive carcinogenesis of AID in diffuse large B-cell lymphoma (DLBCL). The physiological properties of these nanomaterials were examined using transmission electron microscopy (TEM). The inhibition of NPs to AID was primarily identified by the affinity interaction prediction between reactive oxygen species (ROS) and AID through molecular dynamics and molecular docking technology. The cell apoptosis and ROS generation in US-triggered NPs treated DLBCL cells (with high levels of AID) were also detected to indicate the sonosensitivity and toxicity of MWO_4_-PEG NPs to DLBCL cells. The anti-lymphoma studies using DLBCL and AID-deficient DLBCL cell lines indicated a concentration-dependent profile. The synthesized MWO_4_-PEG NPs in this study manifested good sonodynamic inhibitory effects to AID and well treatment for AID-positive hematopoietic cancers.

## 1. Introduction

The emerging noninvasive sonodynamic therapy (SDT) with ultrasound (US) plays a special role in anti-tumor therapy by activating sonosensitizers and generating highly cytotoxic reactive oxygen species (ROS) to induce tumor cell apoptosis. US can penetrate deep soft tissues up to 10 cm due to the frequency beyond human hearing (>20 KHz) and the unique advantage of minimal tissue scattering [1,2,3]. Furthermore, by controlling the frequency, US may precisely target tumor sites while avoiding damage to adjacent normal tissues. Currently reported sonosensitizers mainly include organic and inorganic sonosensitizers [4]. The traditional organic sonosensitizers are usually porphyrin derivatives or other fat-soluble small molecules with poor water solubility, limited ultrasonic stability, detrimental phototoxicity, fast blood clearance, and reduced sonodynamic therapy (SDT) effectiveness [5,6]. Unlike organic materials, inorganic materials are widely reported to feature high physiological, chemical stability and good biocompatibility. These inorganic materials could be used as sonosensitizers to perform efficient tumor treatment. By rationally designing the nanostructure of the semiconductor nanoplatform and controlling its physical and chemical properties, the application of the semiconductor nanoplatform in SDT nanomedicine can be significantly expanded [7,8,9,10]. Recently, transition-metal-based sonosensitizers have been intensively explored because the special d-orbital electron of the transition metal can delay the direct recombination of h+ and e−. Transition-metal-based tungstate has been investigated as a superior nanocatalyst for energy and environmental applications [11,12,13,14]. 

Diffuse large B-cell lymphoma (DLBCL) is the most common form of non-Hodgkin lymphoma, with an incidence of 30~40% [15]. Only 10% of these DLBCL patients can be cured by traditional chemotherapy, and the prognosis of the remaining 90% is still poor [16]. Identifying new key proto-oncoprotein and exploring their inhibitors provide an effective way to treat DLBCL. Activation-induced cytidine deaminase (AID) could cause chromosomal translocations and/or mutations in proto-oncogenes, thus promoting DLBCL formation [17,18,19,20]. Therefore, exploring inhibitors to down-regulate AID might be an efficient approach to treating malignant neoplasms of the hematologic system. 

In this study, we synthesized four tungstate sonosensitizer MWO_4_-PEG nanoparticles (M = Fe Mn Co Ni) as inhibitors to AID to identify the anti-lymphoma effect of these AID inhibitors to treat AID-positive neoplasm. The inhibition was first identified by the interaction between ROS and AID proteins via molecular docking technology. The anti-lymphoma effects of these inhibitors were evaluated by detecting the cell apoptosis and ROS generation of AID-positive DLBCL cells treated by US-triggered NPs. Here, our synthesized inhibitors manifest good inhibition to AID and effective anti-lymphoma therapy (Figure 1).

## 2. Results

### 2.1. Synthesis and Characterization of Small Molecule Inhibitors to AID

Based on effective inorganic materials mediated sonodynamic therapy (SDT) to cancer, we synthesized MWO_4_-PEG nanoparticles (NPs), in which M = Mn, Ni, Fe, Co. As shown in Figure 2, scanning electron microscope (SEM) showed that NiWO_4_-PEG, FeWO_4_-PEG, and CoWO_4_-PEG are nearly spherical with an average diameter of ≈66.5nm, ≈25.5 nm, and ≈65.3nm, respectively. MnWO_4_-PEG is a rod with an average size of ≈70.8 nm in length and ≈18.9 nm in diameter (Figure 2A–D). The TEM images and corresponding energy dispersive spectrometer (EDS) elemental mapping displayed a homogeneous distribution of M, W, O, and C elements in MWO_4_-PEG NPs, which showed that PEG was successfully grafted onto the surface of NPs (Figure 2). These small particles have good solubility and the property of easy entry into cells. Most importantly, these inhibitors show great potential to induce the generation of abundant reactive oxygen species (ROS) (including singlet oxygen [^1^O_2_] and hydroxyl radicals [·OH]) (Figure 2E,F) [1,2,3]. 

### 2.2. Molecular Dynamic Stimulation of the Interaction between ROS and AID

ROS was reported to destroy cell structure and damage many enzymes’ functions [21,22]. So specific inhibition of ROS-producing enzymes is effective in clinical treatment [23]. Based on these concepts and the deleterious role of AID in DLBCL, we tended to simulate the affinity of ROS to AID, thus exploring and synthesizing inhibitors targeting AID. The molecular dynamics and molecular docking results showed that the ·OH had a good affinity with the Ile178 amino acid on AID, and ^1^O_2_ owned well affinity with both Ile178 and Arg179 amino acid on AID (Figure 3B). The Ile178 and Arg179 are precisely located near the functional domain of AID, that is, the nuclear export sequence (NES), which determines the class switch recombination (CSR) function of AID (Figure 3A). The interaction between ROS and AID seriously affected the function of AID. The molecular dynamic stimulation suggested tremendous potential of ROS inhibiting AID.

### 2.3. Application of MWO_4_-PEG NPs to Treat DLBCL

To identify the anti-lymphoma effect of the four MWO_4_-PEG NPs to AID, we detected the cytotoxicity caused by increasing concentration (0, 50, 100, 150, 200 μg/mL) of four NPs. As shown in Figure 4, The upper figures in Figure 4A manifest obvious cell apoptosis detected by flow cytometry (Figure 4A, the upper), and the lower statistical bars in Figure 4A further verified the NPs mediated specific neoplasm cell killing effect (Figure 4A, the lower). Compared with a blank untreated and NPs treated group with administration of different concentrations of NPs, significant cellular damage was found in human OCI-LY19 DLBCL cells treated with or without US irradiation. Notably, even the minimum concentration of the four MWO_4_-PEG NPs (50 μg/mL) caused obvious tumor cell death. When using 200 μg/mL NPs, the tumor cell killing rate was at least six-fold higher than the untreated groups, reaching up to 16-fold, suggesting an outstanding capability in anti-lymphoma effect at the cellular level (Figure 4A). In addition, the ROS generation rate in OCI-LY19 caused by the four NPs with gradient concentration was 8–60 fold higher than the untreated groups, showing an increasing trend similar to that of cell apoptosis rates (Figure 4B). The ROS generation also manifested cellular damages caused by NPs treatment (Figure 4B). And the DLBCL cell-killing effect of MWO_4_-PEG NPs showed a concentration-dependent manner (Figure 4). Collecting the aforementioned data, it is clear that the synthesized PEGylated MWO_4_ NPs show an obvious anti-lymphoma effect. 

### 2.4. Inhibition of MWO_4_-PEG NPs to AID in DLBCL

To further explore the inhibition of NPs to AID, the AID deficiency (AIDKO) SU-DHL-4 DLBCL cells were generated by the CRISPR/Cas9 technique targeting the human AICDA gene, which encodes AID protein [24,25]. After being treated by increasing the concentration (0, 50, 100, 150, 200 μg/mL) of the four NPs, the apoptosis rate and ROS generation of SU-DHL-4 and AIDKO SU-DHL-4 cells were detected by flow cytometry. Both NPs treatments could cause numerous cell apoptosis, and the apoptosis rate (PI^+^) increased with the elevated concentration of NPs with or without US activation (Figure 5A,B). In addition, AID deficiency contributed to the killing effect of the NPs on DLBCL cells (Figure 5A,B). The inhibition effect of NPs to AID combined with AID deficiency mediates the most effective cell apoptosis than NPs alone and AID deficiency alone, and the cell apoptosis increased by 2–20 fold in AID-deficient DLBCL cells (Figure 5A,B). Moreover, the measurement of ROS generation further identifies the anti-lymphoma function of the NPs. The AID deficiency combined with NPs treatment caused the most obvious ROS-mediated cellular damage. ROS generation increased by at least 1.5-fold, reaching 20-fold in AID-deficient DLBCL cells (Figure 5C,D). Collecting these data indicated that the Us-triggered NPs show apparent cytotoxicity to DLBCL cells through targeting the tumor-promoting factor AID. 

The AID levels in DLBCL cells with the administration of four NPs (200 μg/mL) were detected to identify the inhibition of four NPs to AID. The immunoblotting showed that the NPs manifested a remarkable inhibition of AID expression in OCI-LY19 DLBCL cells (Figure 6A). The gray density of the immunoblotting bands was analyzed, showing obvious inhibition of AID by four NPs in DLBCL cells (Figure 6B). Also, for NPs treated with SU-DHL-4 and AIDKO SU-DHL-4 cells (Figure 6C,D), the NPs caused obvious down-regulation of AID compared with the untreated group. Furthermore, NPs administration to AIDKO SU-DHL-4 DLBCL cells almost wiped out all AID expressions by comparing with the internal protein GAPDH. The analysis for gray density of the immunoblotting bands indicates that apparent AID inhibition in DLBCL cells by the treatment with four inhibitors and AID deficiency enhanced the inhibition (Figure 6E). These results verify the remarkable inhibition of four NPs to AID and indicate the NPs combined with AID deficiency enlarged the anti-lymphoma effect on DLBCL. 

### 2.5. Sonosensitivity and Toxicity of MWO_4_-PEG NPs

The sonosensitivity and toxicity of the four inhibitors were detected to compare their compatibility in SDT to DLBCL. As shown in Figure 7, US activation to four NPs induced a slight elevation in cell death rate (PI^+^ cells), while adding the factor of AID deficiency, the US triggering cell apoptosis increased by more than 30-fold (Figure 7C). In addition, the oxidant-mediated anti-lymphoma of the four NPs were compared in SDT to DLBCL. With the treatment with an elevated concentration of four NPs, US-induced ROS generation achieved more than a 60-fold increase compared to no US activation in OCI-LY19 (Figure 8A). Also, US activation induced ROS generation in SU-DHL-4 cells more than 20-fold than in cells without US irradiation. In contrast, the ROS generation in AID-deficient DLBCL cells treated by CoWO_4_–PEG in the presence of US manifests nearly more than 20-fold than no US activation. Collectively, these result comparisons from the detection of cell apoptosis and ROS generation between US and no US activation suggest a well sonosensitivity of MWO_4_-PEG NPs.

Moreover, the cell apoptosis detection results showed that the PEGylated CoWO_4_ mediated the best cell-killing effect among the four NPs (Figure 7B). Also, AID deficiency enlarged this disparity (Figure 7C). In the measurement of ROS, the results also showed that PEGylated CoWO_4_ mediated the most ROS generation among the four NPs (Figure 8A,B) and more ROS generation in AID-deficient DLBCL cells. In the immunoblots for detecting AID expression, it is evident that US-triggered PEGylated CoWO_4_ manifested as more than a 400-fold increase of inhibition effect to AID in AID-deficient DLBCL cells (Figure 6). These data strongly indicate that CoWO_4_-PEG NPs possess the best sonosensitivity to AID inhibition and the best toxicity to DLBCL cells in a concentration-dependent manner.

Interestingly, the cell apoptosis and ROS generation with or without US triggering symbolized an opposite trend in DLBCL cells treated with NPs. This may be a hint at the interaction between ROS and AID. The more ROS generation, the more interaction between ROS and AID, and the more AID consumption in DLBCL cells. Finally, the carcinogenesis of AID in DLBCL was neutralized or even eliminated by our synthesized NPs, manifesting an increase in cell apoptosis. This abundant ROS-AID interaction confirms a cue that these synthesized NPs could be used as specific inhibitors to AID in the clinical treatment of AID-positive neoplasms.

## 3. Discussion

In developing efficient, accurate, non-invasive, or minimally invasive cancer treatment [26,27,28], photodynamic therapy (PDT) attracted great attention because of its noninvasive properties. However, PDT faces the intrinsic critical issues of the low tissue-penetrating depth of laser and significant phototoxicity [29,30,31]. As another rising typical noninvasive irradiation source, ultrasound (US) could activate sonosensitizers and induce the generation of reactive oxygen species (ROS) to induce tumors [1,2,3,4]. Unlike the deficiency of low tissue-penetrating depth in PDT, US can penetrate deep soft tissues. In addition, US may precisely target tumor sites while avoiding damage to adjacent normal tissues by changing the frequency [1,2,3,4]. Here, the synthesized inorganic PEGylated MWO_4_ nanoparticles (NPs) (M = Fe, Co, Mn, Ni) show stable physiological and chemical features and good biocompatibility. In addition, based on well characterizing these NPs by scanning electron microscope (TEM) and energy dispersive spectrometer (EDS) elemental mapping, the different sizes of the four NPs showed no noticeable impact on physicochemical properties and very slight influence on the effect of anti-lymphoma therapy. Comparing the cell apoptosis and ROS generation of diffuse large B cell lymphoma (DLBCL) cells treated by NPs in the absence or presence of US irradiation, US-triggering DLBCL cell apoptosis, and ROS generation indicated dozens of times more efficient than that of no US activation. This predominantly manifests the high performance of sonodynamic therapy (SDT) of PEGylated MWO_4_ NPs to DLBCL.

Many studies reported that AID promotes DLBCL progression [17,18,19,20]. Numerous reports focus on the mechanism of AID-mediated gene instability and try to use gene therapy targeting AID to treat AID-associated diseases and cancers [17,18,19,20]. However, compared with the complexity and difficulty of gene targeting therapy, directly using drugs to down-regulate AID might be a more efficient approach to treat malignant neoplasms of the hematologic system. Although many drugs act as inhibitors to AID, thus aiming to suppress the hematologic malignancies and tumors, these drugs inhibit AID function indirectly without immediate work [23,24,25,32,33]. There are no inhibitors directly targeting AID that have been developed at present. In this study, we successfully synthesized the high-performance MWO_4_-PEG NPs [28,29,30]. These inhibitors detected protein levels and significantly decreased AID protein expression. All NPs showed apparent and quick AID inhibition, which verified the powerful inhibition of PEGylated MWO_4_ NPs to AID. It allows for the clinical application of AID inhibitors in treating DLBCL.

The data in this study indicated that CoWO_4_-PEG NPs induced the best AID elimination among the four NPs in AID-deficient DLBCL cells. Moreover, the sonosensitivity and toxicity of PEGylated CoWO_4_ NPs to DLBCL cells showed the best effect among the four kinds of NPs. As is reported, Mn^2+^, Ni^2+^, Fe^2+,^ and Co^2+^ were all used in the treatment of cancers mainly through destroying structural support, damaging enzymatic cofactors, or inducing dysfunction of normal electron transportation of the cancer cells [34,35]. In addition to these mechanisms, Co^2+^ usually kills cancer cells by causing hypoxia, which can be fatal to cancer cells that need oxygen to survive and proliferate. Compared with the low-effect damages to cancers caused by Mn^2+^, Ni^2+^, and Fe^2+^, the immediate hypoxia effect of Co^2+^ on cancer cells may be more efficient. This could be the most reasonable reason to explain the best effect of PEGylated CoWO_4_ NPs in treating DLBCLs [36,37].

It was found that the US irradiation induced ROS generation in DLBCL cells decreased while cell apoptosis increased. This could exactly explain to be the mechanism that ROS interacted with AID to neutralize the deleterious effect in tumor formation. The NPs induced abundant ROS could interact with a large amount of AID in DLBCL cells, and the decrease of AID level mediated a large extent of DLBCL cell apoptosis. The rich depletion of AID by ROS was achieved by the interaction between singlet oxygen [^1^O_2_], hydroxyl radicals [·OH], and AID. It further confirms the specific inhibition of four MWO_4_-PEG NPs to AID. To realize the exploration of specific inhibitors to AID is the exact aim of our study, CoWO_4_-PEG NPs synthesized here would be a good choice.

It is highly expected that the development of new types of high-performance multifunctional AID inhibitors shown in this study would provide efficient SDT to cancer through inhibiting AID and could be widely used in treating AID-positive neoplasm and tumors with advantages of cost-effectiveness, convenience, and noninvasiveness.

## 4. Conclusions

AID is reported to promote neoplasm and solid tumors in many studies. Thus, deriving AID inhibitors to explore the anti-tumor effect is an effective way of treating neoplasms and tumors. Inspired by the application of sonodynamic therapy (SDT) triggered by ultrasound (US) in anti-tumor therapy, four sonosensitizer MWO_4_-PEG NPs (M = Fe Mn Co Ni) were synthesized in this study and taken as AID inhibitors to treat DLBCLs. These inhibitors exhibited efficient reactive oxygen species (ROS) generation capability, and they mediated and increased DLBCL cell apoptosis rate with US irradiation. By detecting protein levels, these inhibitors induced a significant decrease in AID protein expression. The apparent inhibition of MWO_4_-PEG NPs to AID and cell-killing effect indicated good sonosensitivity and toxicity of NPs to DLBCLs. This is evident of the good anti-lymphoma properties of the sonosensitizer in SDT to DLBCLs. It is highly expected that the development of new types of high-performance multifunctional AID inhibitors shown in this study would provide efficient US-triggered cancer therapy through inhibiting AID and could be widely used in treating AID-associated neoplasms and tumors with the advantages of cost-effectiveness, convenience, and noninvasiveness.

## 5. Materials and Methods

### 5.1. Materials

Iron(II) chloride tetrahydrate (FeCl_2_·4H_2_O), cobalt chloride hexahydrate (CoCl_2_·6H_2_O), manganese chloride tetrahydrate (MnCl_2_·4H_2_O), nickel chloride hexahydrate (NiCl_2_·6H_2_O), Sodium tungstate dihydrate (Na_2_WO_4_·2H_2_O), hexadecyl trimethyl ammonium Bromide (CTAB), 1,3-diphenylisobenzofuran (DPBF) and 3,3,5,5-Tetramethylbenzidine (TMB) were purchased from Shanghai Macklin Biochemical Co., Ltd. (Shanghai, China). All chemicals were of analytical grade and used without further purification.

The ROS detection kit (S0033S) and propidium iodide (PI) (ST511) were purchased from Beyotime Biotechnology Co., Ltd. (Shanghai, China). The primary antibodies, Anti-GAPDH and anti-AID, were purchased from Abways Technology Co., Ltd. (Shanghai, China). and North American ImmunoWay Biotechnology Company (TX, USA). The ECL Substrate Kit (P0018FS) was obtained from Beyotime Biotechnology Co., Ltd. (Shanghai, China). 

Human DLBCL SU-DHL-4 and OCI-LY19 cell lines were purchased from BeNa Culture Collection (BeNa, #BNCC340176, #BNCC338225) (Suzhou, China). 

### 5.2. Cell Culture and Construction of Stable Clones

AID deficiency (AIDKO) SU-DHL-4 cells were generated by CRISPR/Cas9 technique targeting the *AICDA* gene. The details were described in our previous study [20,21]. Cells were cultured in IMDM (Hyclone) supplemented with 10% fetal bovine serum (FBS) (Sigma), non-essential amino acids, and penicillin—streptomycin (1%), and β-mercaptoethanol (50 μM). All cells were incubated at 37 °C in a humidified incubator containing 5% CO_2_ (Thermo Fisher Scientific, MA, USA). 

### 5.3. Molecular Dynamics

The three-dimensional structure of the AID protein was modeled based on the amino acid sequence. The chosen compounds were prepared through hydrogens addition, calculation of partial charges, and energy minimization using CHARMM Force Field at thoroughness = 0.01. The best poses and binding score values of the selected compounds were detected by applying procedures and parameters. The ROS were set as ligands, and AID was set as the receptor. The binding site was set as a sphere with a 40 Å radius to ensure that the entire protein structure was included. Ligand structures were constructed and optimized using the Materials Studio. Docking simulations of the small molecules ROS and AID were performed using the CDOCKER program with the default parameters. Moreover, the obtained docking poses of the tested compounds were assessed according to the binding energy scores; RMSD values; conformity of the binding forces with the co-crystalized ligand, celecoxib. Finally, the best pose was isolated, placed in the active site, and saved as a picture to export as a JPEG file.

### 5.4. Synthesis of MWO_4_ (M = Fe Mn Co Ni) Nanoparticles

MWO_4_ (M = Fe Mn Co Ni) nanoparticles were synthesized at room temperature. In detail, CTAB (0.365 g) was scattered in the NiCl_2_·6H_2_O solution (0.02 M, 50 mL) via vigorous magnetic stirring until the solution cleared. The above solution was mixed with 50 mL Na_2_WO_4_·2H_2_O solution (0.02 M) under an ultrasonic cell disrupter for 30 min. Intervals of 15 s are required for every 5 s of ultrasound, and the temperature of the solution must be below 323 K. The sediment was centrifuged and washed repeatedly with cyclohexane and ethanol and then dried at 50 °C for 24 h. FeWO_4_, CoWO_4_, and MnWO_4_ nanoparticles can be obtained by the same method.

To conjugate PEG, MWO_4_ was mixed with NH_2_−PEG2000-COOH (1:1 by weight), and the reaction proceeded under stirring at 25 °C for 24 h. At last, MWO_4_ nanoparticles were washed with DI water prior to any use. The size was determined by transmission electron microscope (TEM) imaging (JEM-2100 TEM) (JEOL, Beijing, China). The synthesis of MWO_4_ was repeated at least three times, and the average diameter of MWO_4_-PEG NPs could be controlled very well.

### 5.5. Measurement of ROS Generation

OCI-LY19, SU-DHL-4, and AIDKO SU-DHL-4 cells were incubated with various concentrations of the four types of MWO_4_-PEG nanoparticles (NPs) (0, 50, 100, 150, 200 µg/mL) for 8 h, followed without or with US irradiation (40 kHz, 3 W/cm^2^) for 2 min. All cells were harvested after another incubation for 1 h [38]. 

For ROS detection, the treated OCI-LY19 and SU-DHL-4 cells were stained with DCFH-DA (20 μM) for 30 min under 37 °C. Then the ROS^+^ cells were detected by a CytoFLEX Flow Cytometer (Beckman Coulter, CA, USA), and a total of 1 × 10^4^ events were collected [39]. 

### 5.6. Detection of Cell Apoptosis

OCI-LY19, SU-DHL-4, and AIDKO SU-DHL-4 cells were treated by increasing concentration of four NPs (0, 50, 100, 150, 200 µg/mL) for 8 h, followed without or with US irradiation (40 kHz, 3 W/cm^2^) for 2 min. All cells were harvested after another incubation for 1 h. A flow cytometer was used to quantitatively evaluate the number of dead cells. 

For apoptosis rate detection, the cells were treated with four types of MWO_4_-PEG NPs and then stained with propidium iodide (20 μM) for 30 min at room temperature. The PI^+^ cells were detected by a CytoFLEX Flow Cytometer (Beckman Coulter, CA, USA), and a total of 1 × 10^4^ events were collected. 

### 5.7. Western Blotting

OCI-LY19, SU-DHL-4, and AIDKO SU-DHL-4 cells in a 12-well plate (5 × 10^4^ per well) were exposed to the four types of MWO_4_-PEG NPs (200 µg/mL) for 8 h, followed by US irradiation (40 kHz, 3 W/cm^2^) for 2 min. All of the cells were harvested after another incubation for 1 h. Next, the cells were lysed by RIPA lysis buffer, and the amount of extracted protein content was quantified through BCA Protein Assay Kit. The lysates were then loaded in 10% sodium dodecyl sulfate-polyacrylamide (SDS-PAGE) gel, which was further transferred to a poly (vinylidene difluoride) (PVDF) membrane. After blocking the membrane with skim milk (5% *w/v*) in Tris buffer saline with Tween 20 (TBST) for 1 h, the membrane was thoroughly washed with TBST. Immunoblotting was performed by incubating the protein with anti-AID (YT5566, 1:1000 dilution) and anti-GAPDH (AB0036, 1:1000 dilution) at 4 °C overnight. Afterward, the membrane was further treated by anti-Rabbit HRP secondary antibody for 2 h and examined via Western-Ready enhanced chemiluminescence (ECL) Substrate Kit [24,25]. 

### 5.8. Statistical Analysis 

Two-way ANOVA multiple tests were performed with GraphPad Prism 6.0 (GraphPad Software, LaJolla, CA, USA). Data were considered statistically significant if *p* values were less than 0.05, as indicated.

## Figures and Tables

**Figure 1 molecules-27-07143-f001:**
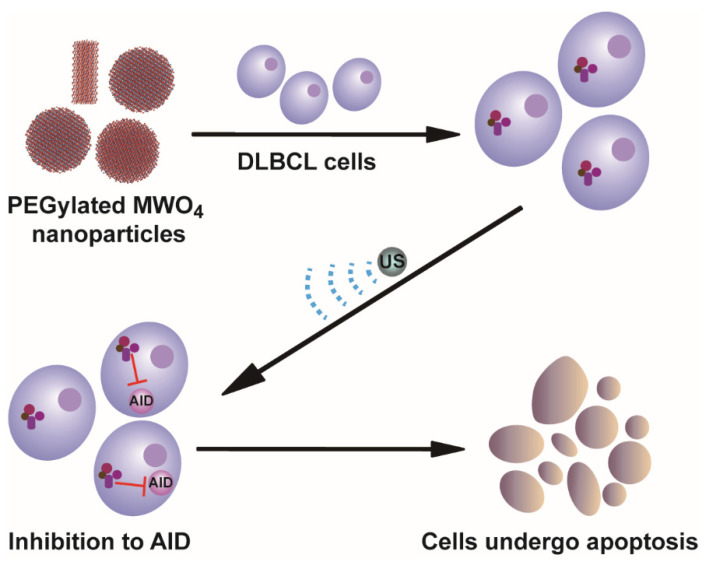
Sonodynamic therapy with the synthetic AID inhibitors (PEGylated MWO_4_) to AID-positive lymphoma.

**Figure 2 molecules-27-07143-f002:**
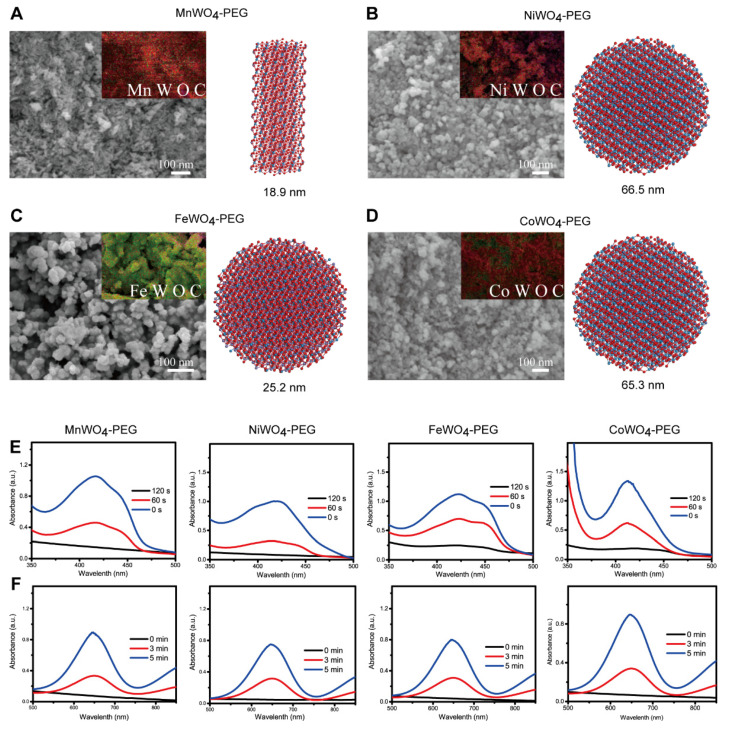
Characterization of MWO_4_-PEG NPs. The SEM images and the EDS images (the left), crystal structure and size (the right) of MnWO_4_-PEG NPs (**A**), NiWO_4_-PEG NPs (**B**), FeWO_4_-PEG NPs (**C**) and CoWO_4_-PEG NPs (**D**). The Time-dependent oxidation of DPBF indicates ^1^O_2_ generation by MWO4-PEG under US irradiation (**E**), and the Time-dependent oxidation of TMB due to •OH generation by MWO4-PEG under US irradiation (**F**).

**Figure 3 molecules-27-07143-f003:**
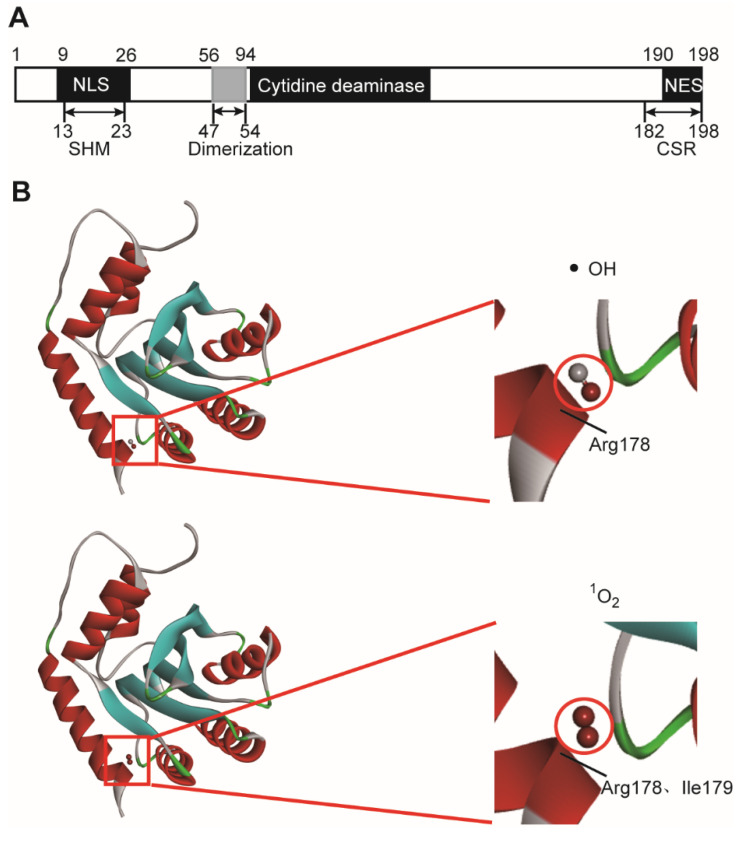
The affinity between AID and ROS (including ·OH and ^1^O_2_). (**A**) The primary structure of AID protein with including the functional domains. SHM, somatic hypermutation; CSR, class switch recombination. NLS, nuclear localization sequence; NES, nuclear export sequence. (**B**) Affinity simulation between singlet oxygen (^1^O_2_) or hydroxyl radicals (·OH) and AID was performed by the CDOCKER program. AID and ROS structures were constructed and optimized using Materials Studio. The red square fields were enlarged and are displayed on the right, the circled drugs indicated OH or ^1^O_2_, red indicates O, and gray indicates H. Arg178 denotes that the 178 amino acid on AID is Arginine; Ile179 denotes that the 178 amino acid on AID is Isoleucine.

**Figure 4 molecules-27-07143-f004:**
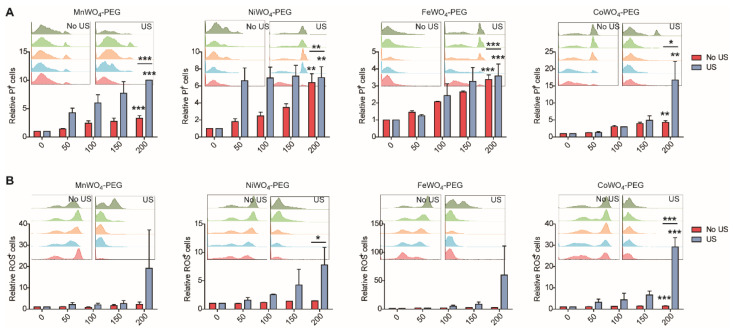
MWO_4_-PEG NPs mediated cytotoxicity to OCI-LY19 cells. (**A**) The relative apoptosis rate of human OCI-LY19 cells treated by increasing concentration (0, 50, 100, 150, 200 μg/mL) of four MWO_4_-PEG NPs with or without US irradiation. The propidium iodide (PI) was used to stain the NPs treated OCI-LY19 cells to indicate apoptosis cells. The percentage of PI^+^ cells after administration of MWO_4_-PEG NPs with or without US activation was detected by flow cytometry. (**B**) The percentage of ROS generation by OCI-LY19 DLBCL cells after incubation with different concentrations (0, 50, 100, 150, 200 μg/mL) of MWO_4_-PEG NPs in the presence or absence of US. The upper showed the primary data obtained from flow cytometry, and the lower showed the statistical chart of relative PI^+^ or ROS^+^ cells. The data shown are representative of 3 independent experiments. Data are represented as mean ± SD. *, ** and *** represent *p* < 0.05, *p* < 0.01 and *p* < 0.001, respectively.

**Figure 5 molecules-27-07143-f005:**
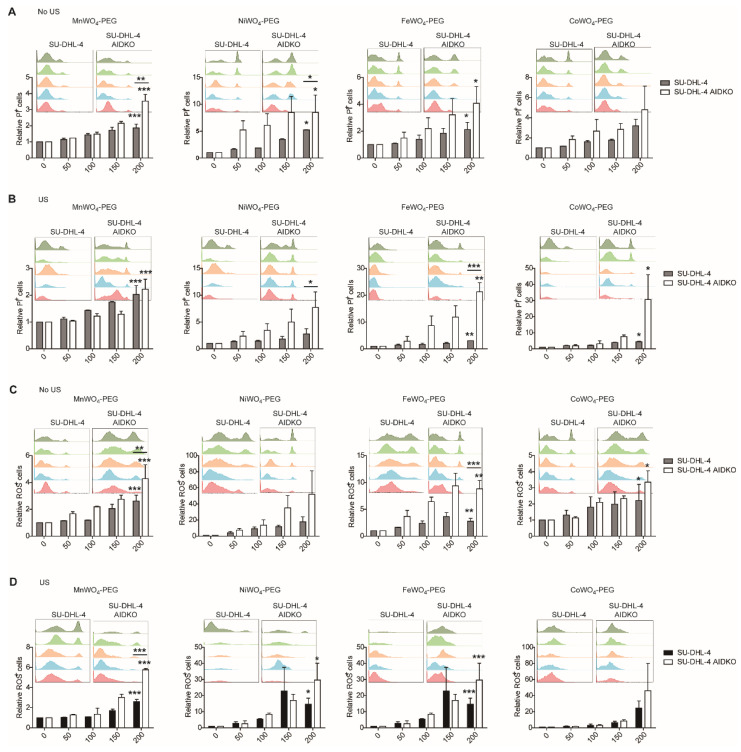
Cytotoxicity to DLBCL caused by MWO_4_-PEG NPs combined with AID deficiency. (**A**) The relative PI^+^ generation of human SU-DHL-4 and AIDKO SU-DHL-4 cells was treated by increasing concentration (0, 50, 100, 150, 200 μg/mL) of four MWO_4_-PEG NPs without US irradiation. The PI was used to stain the NPs treated DLBCL cells to indicate apoptosis cells. The percentage of PI^+^ cells after the administration of MnWO_4_-PEG NPs without US activation was detected by flow cytometry. (**B**) The apoptosis rate of human SU-DHL-4 and AIDKO SU-DHL-4 cells was treated by increasing concentration (0, 50, 100, 150, 200 μg/mL) of four MWO_4_-PEG NPs with US irradiation. PI was used to stain the NPs treated DLBCL cells to indicate apoptosis cells. (**C**) The percentage of ROS generation by SU-DHL-4 and AIDKO SU-DHL-4 cells after incubation with different concentrations (0, 50, 100, 150, 200 μg/mL) of MWO_4_-PEG NPs in the absence of US. (**D**) The percentage of ROS generation by SU-DHL-4 and AIDKO SU-DHL-4 cells after incubation with different concentrations (0, 50, 100, 150, 200 μg/mL) of MWO_4_-PEG NPs in the presence of US. The upper showed the primary data obtained from flow cytometry. The lower showed the statistical chart of relative PI^+^ or ROS^+^ cells. The data shown are representative of 3 independent experiments. Data are represented as mean ± SD. *, ** and *** represent *p* < 0.05, *p* < 0.01 and *p* < 0.001, respectively.

**Figure 6 molecules-27-07143-f006:**
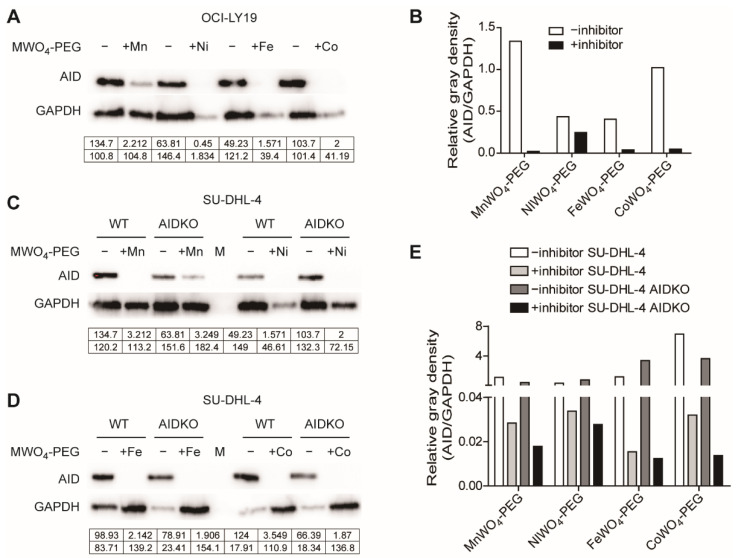
Inhibition of MWO_4_-PEG NPs to AID in DLBCL. (**A**) The AID and GAPDH protein level of OCI-LY19 DLBCL cells with the treatment of four NPs with the concentration of 200 μg/mL in the presence of US irradiation. The gray density was indicated under these bands correspondingly. (**B**) The bars showed the relative gray density of AID to internal protein GAPDH in inhibitor-treated OCI-LY19 cells. (**C**) The AID and GAPDH protein level of SU-DHL-4 and SU-DHL-4 AIDKO DLBCL cells with the treatment of MnWO4-PEG NPs and NiWO_4_-PEG NPs with the concentration of 200 μg/mL with US activation. The gray density was indicated under these bands correspondingly. (**D**) The AID and GAPDH protein levels of SU-DHL-4 and SU-DHL-4 AIDKO DLBCL cells with the treatment of FeWO_4_-PEG NPs and CoWO_4_-PEG NPs with the concentration of 200 μg/mL. The gray density was indicated under these bands correspondingly. (**E**) The bars showed the relative gray density of AID to internal protein GAPDH in NPs treated with SU-DHL-4 and AIDKO SU-DHL-4 cells.

**Figure 7 molecules-27-07143-f007:**
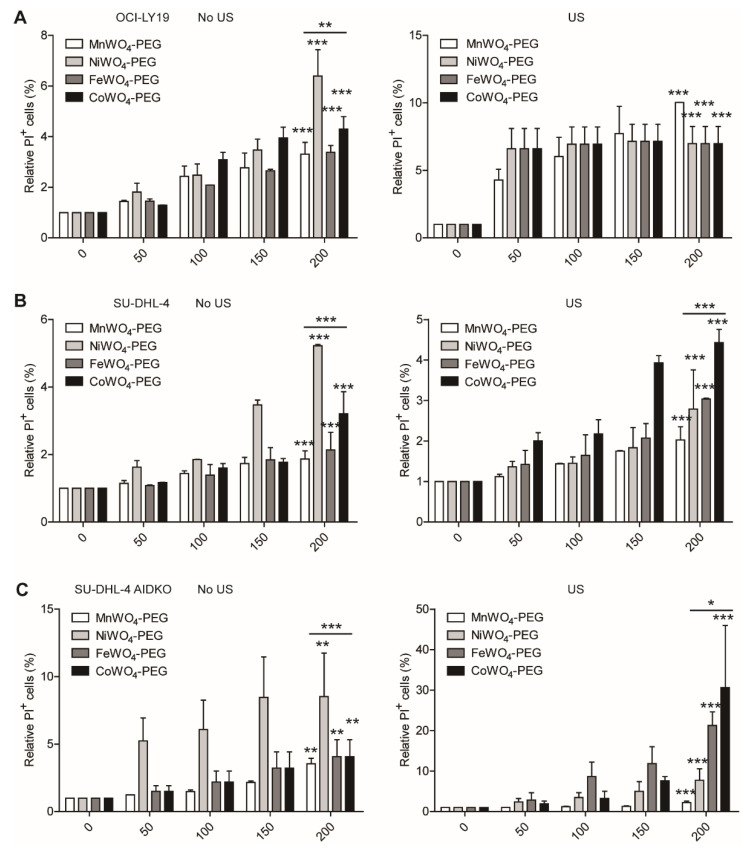
US-triggered toxicity of MWO_4_-PEG NPs. The relative apoptosis rate of human OCI-LY19 (**A**), SU-DHL-4 (**B**), and AIDKO SU-DHL-4 (**C**) cells treated by increasing concentration (0, 50, 100, 150, 200 μg/mL) of four MWO_4_-PEG NPs with (left figure) or without (right figure) US irradiation. The propidium iodide (PI) was used to stain the NPs treated DLBCL cells to indicate apoptosis cells. The percentage of PI^+^ cells was detected by flow cytometry. The data shown are representative of 3 independent experiments. Data are represented as mean ± SD. *, ** and *** represent *p* < 0.05, *p* < 0.01 and *p* < 0.001, respectively.

**Figure 8 molecules-27-07143-f008:**
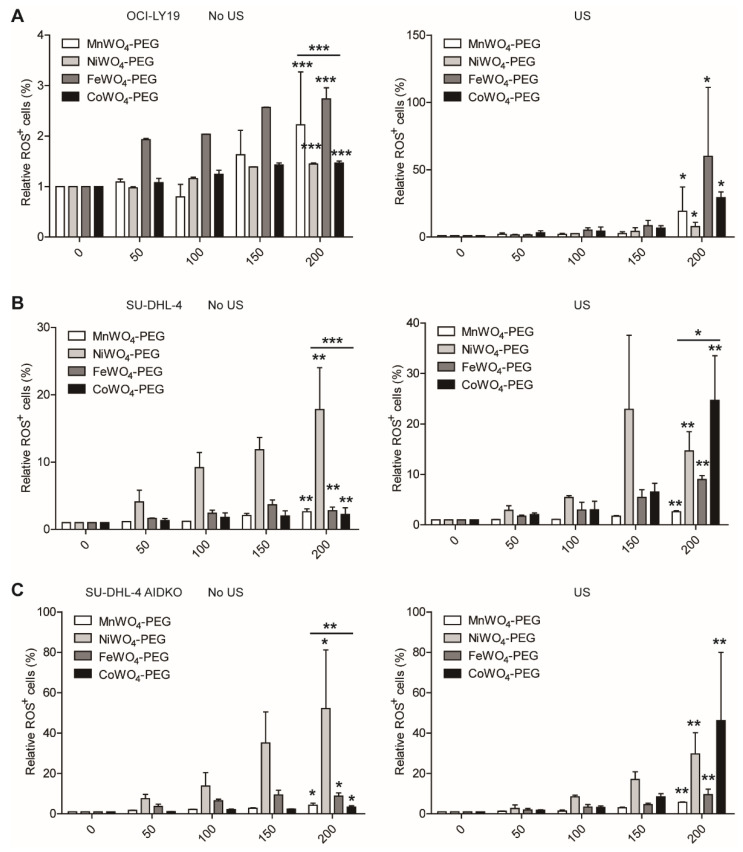
US-triggered toxicity of MWO_4_-PEG NPs. The relative ROS^+^ generation of human OCI-LY19 (**A**), SU-DHL-4 (**B**), and AIDKO SU-DHL-4 (**C**) cells treated by increasing concentration (0, 50, 100, 150, 200 μg/mL) of four MWO_4_-PEG NPs with (left figure) or without (right figure) US irradiation. The ROS was used to stain the NPs treated DLBCL cells to indicate ROS generation. The percentage of ROS^+^ cells was detected by flow cytometry. The data shown are representative of 3 independent experiments. Data are represented as mean ± SD. *, ** and *** represent *p* < 0.05, *p* < 0.01 and *p* < 0.001, respectively.

## Data Availability

Not applicable.

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
