# Peer review of "Synthesis and Biological Evaluation of PEGylated MWO4 Nanoparticles as Sonodynamic AID Inhibitors in Treating Diffuse Large B-Cell Lymphoma"

_molecules, 2022, doi:10.3390/molecules27217143_

Round 1

Reviewer 1 Report (Previous Reviewer 1)

It was a manuscript about the fabrication and utilization of four different nanoparticles as inhibitors to activation-induced cytidine deaminase. Here are some comments on this study that should be considered before publication:

·       Please improve the quality of the abstract.

·       Please add the manufacturer country of the used instrument.

·       Please add more characterization tests to confirm the correct preparation of nanoparticles.

·       Please add references related to sections 4.5 to 4.7.

·       Please compare the effect of using ultrasound on ROS production inside cells.

·       How do you confirm the specificity of the fabricated nanoparticles for cancer cells?

·       Please add the conclusion part to the main text.

Author Response

Reviewer 2 Report (New Reviewer)

Author Response

Reviewer 3 Report (New Reviewer)

The manuscript Synthesis and Biological Evaluation of PEGylated MWO4 Na-2 noparticles as Sonodynamic AID Inhibitors in Treating Diffuse 3 Large B-cell Lymphoma” deals with the synthesis of four MWO4-PEG nanoparticles and the application of them as sonosensitizers and inhibitors of induced cytidine deaminase (AID).

The manuscript is not well written and is not of easy reading. In my opinion, I can’t recommend this work for pubblication. The character of the paper is not so high to fit to the standard of the journal. I apologize, but in my opinion this paper needs major revisions and improvements before publishable in this journal. In this version, the manuscript has to be rejected. The fundamental point is that the English should be improved. The authors should revise the entire paper because there are a number of errors, a number of not precise expressions, phrases without sense. I can’t report all of them, because they are scattered on all the paper, and make the lecture impossible. For example:

lines 38-44: the sentences are not correct. A point is missed at line 41 and then the phrases have no sense with repetitions.

Line 52: demonstrated should be corrected

Line 57-58: “with relapsed and refractory”: this expression has no sense

Line 68-69: a noun is missed: effects?

Lines 71-72: the phrases are not correct

Figure 1: not “cells go to apoptosis”, but cells undergo apoptosis”

Line 86-87: did the authors really want to use “entitled” and “modulate”? which is the sense?

Line 99-100: the phrase has no sense

Paragraph 2.2: the authors always write “stimulation” instead of “simulation”

Line 144: eitout should be corrected

And more and more.

The remarks listed below must be taken into account even if the paper is resubmitted to this or another journal.

The authors consider as ROS, singlet oxygen and hydroxyl radicals. However, they don’t report any evaluation of the identification of ROS induced by their method and not distinguish the effects of the two cited. Moreover, the discussion of the results of molecular dynamic simulation and docking to identify the residues that interact with the two ROS molecules, appears very weak. Do the authors think that the first (and unique) parameter for this kind of interaction is the steric hindrance?

The question of interaction of the NPs and serum albumin or other serum proteins was not addressed. However, the authors should study this topic (protein corona) because it is relevant in term of availability of their “drugs”. Moreover, which is the nanotoxicity of these NPs against liver, brain, lung, and kidney?

Minor points:

Figure 4 and 5 are too small

The comparison between Figure 7 and 8 is difficult, the authors should organize better the results.

Round 2

Reviewer 1 Report (Previous Reviewer 1)

-

Author Response

Reviewer 2 Report (New Reviewer)

The resubmission of this manuscript has improved from the previous form. Authors made major revisions and responded to the points highlighted, according to the reviewers’ suggestions.

As minor corrections, I advise authors to include responses to points 2 and 3 in the manuscript as part of section 5.4. Synthesis of MWO4 (M=Fe Mn Co Ni) (point 2) and section 3. Discussion (point 3).

In addition, there are still sentences that need to be improved and corrected concerning English grammar.

After these minor corrections are made, I consider it suitable for publication in Molecules.

Author Response

Reviewer 3 Report (New Reviewer)

the authors revised the manuscript in line with comments and suggested revisions

Author Response

This manuscript is a resubmission of an earlier submission. The following is a list of the peer review reports and author responses from that submission.

Round 1

Reviewer 1 Report

It was a study about the synthesis and evaluation of different types of MWO4-PEG nanoparticles as AID inhibitors for the aim of diffuse large B-cell lymphoma therapy. Despite efforts done for the preparation of this study, it is not suitable to be published in this journal. Here are some comments which could help authors to improve the quality of their manuscript: 

1-      The quality of the abstract is low. Please rewrite it. The same for the introduction section.

2-      There are several grammatical mistakes in the text that should be corrected.

3-      Figure 1 is not good, please replace it with an eye-catching figure.

4-      Please add the subheading “materials” to section 4 and mention the used materials, their purity, and the manufacturer country in it.

5-      Please introduce all the abbreviations at the first-time usage.

6-      How do you confirm the correct preparation of nanoparticles? TEM alone is not sufficient for that.

7-      What is the relationship between the molecular stimulation section and the experimental part? Why did you use drugs for the stimulating part and nanoparticles for the experimental section?

8-      Please add the results of elemental analysis to figure 5.

9-      “Notably, even the minimum concentration of drug treatment (50 μg/ml) caused obvious tumor cell death” which drug?

10-   Please compare the toxicity results of different nanoparticles.

11-   The discussion part is not good. It is like the conclusion, not the discussion. Please rewrite the discussion and add the conclusion.

Reviewer 2 Report

In sufficient details in most aspects, and the parts of the work seem to be disparate and not joined. There are issues with the flow and logical thought.

There are insufficient details on the molecular dynamics, how were they performed. How were the docking performed. What were the key results which were not really shown. I assume blind docking was performed, and the clustering analysis etc should be shown to show where the pockets etc are.

It is also not clear the link to the MWO4 generation from the MD and docking.

The discussion lacks depth and elaboration to the purpose of the work and what they mean.

The raw blots provided do not show the markers and do not have labels and therefore do not have much meaning in themselves.

What brand of the flow cytmeter was used, how many events etc? what were the flurophores used apart from PI? A lot of details not being shown in M&Ms.

Not really apparent how the characterization of the MWO4 using TEM means anything.

A proper revision is required for a re-read.

Minor:

Some proofreading for english e.g. Line 85 should be "were analyzed"

Line 90, degree symbol would be better.

Figures e.g. Figure 3 low resolution.

The graphs in e.g. Figure 6 and 7 should try and show statistical significance if any.

For the benefit of a doubt, recommend major revision for a relook at the manuscript with the issues addressed.